# Endometrial Microbial Profile in Infertile Women with Chronic Endometritis: Intensified Culturomics and 16S rDNA Gene Sequencing

**DOI:** 10.3390/genes16121432

**Published:** 2025-12-01

**Authors:** Bárbara Lara-del-Río, Rocío Sánchez-Ruiz, Mónica Bernal-Sánchez, Francisco Ruiz-Cabello-Osuna, José María Navarro-Marí, José Gutiérrez-Fernández

**Affiliations:** 1Department of Laboratory and Immunology, Hospital Universitario Virgen de las Nieves, 18014 Granada, Spain; 2Ph.D. Program in Clinical Medicine and Public Health, University of Granada, 18011 Granada, Spain; 3Department of Gynecology and Obstetrics, Hospital Universitario Virgen de las Nieves, 18014 Granada, Spain; 4Instituto de Investigación Biosanitaria de Granada (ibs.GRANADA), 18012 Granada, Spain; 5Department of Microbiology, Hospital Universitario Virgen de las Nieves, 18014 Granada, Spain; josem.navarro.sspa@juntadeandalucia.es; 6Department of Microbiology, School of Medicine, University of Granada, 18014 Granada, Spain

**Keywords:** chronic endometritis, endometrial microbiota, infertility, next-generation sequencing, culturomics, *Lactobacillus*, *Gardnerella*

## Abstract

Background/Objectives: Chronic endometritis (CE) has been linked to implantation failure, recurrent pregnancy loss, and endometrial dysbiosis with low *Lactobacillus* abundance. We assessed the endometrial microbiota in infertile women with CE and the added value of combining intensified culturomics with 16S rDNA NGS. Methods: Cross-sectional study with descriptive results of 15 endometrial biopsies analyzed in parallel by intensified culture and 16S rDNA (V3–V6) sequencing (RA threshold ≥ 5%). Results: Culture yielded growth in 13/15 samples (86.7%), whereas NGS reported findings in 15/15 (100%). NGS provided additional taxa beyond culture in 14/15 (93.3%), while culture identified taxa missed by NGS in 10/15 (66.7%). In both culture-negative cases, NGS detected ≥1 taxon. *Lactobacillus* spp. appeared exclusively by NGS in 26.7% of samples; *Fannyhessea vaginae* showed the highest mean RA by NGS and did not grow in culture, underscoring complementarity. Conclusions: 16S NGS complements intensified culturomics for characterizing the endometrial microbiota in CE, enhancing detection—especially when culture is negative—and supports a combined, clinically contextualized interpretation. Larger, controlled cohorts are warranted to validate diagnostic and prognostic utility.

## 1. Introduction

Chronic endometritis (CE) is characterized by persistent inflammation of the endometrial mucosa. Histologically, it is defined by the presence of CD138^+^ plasma cells in endometrial stroma and has been associated with infertility, recurrent implantation failure (RIF), and recurrent pregnancy loss (RPL). Although its epidemiology is heterogeneous owing to variability in diagnostic criteria and detection techniques, recent reviews estimate a prevalence of approximately 2.8–56.8% in women with infertility, 14–67.5% in RIF, and 9.3–67.6% in RPL, underscoring its potential clinical relevance. This wide dispersion is explained, in part, by diagnostic heterogeneity: (i) use of hysteroscopy based on non-uniform visual criteria (micropolyps, edema, hyperemia) with variable accuracy; (ii) conventional histology versus CD138 immunohistochemistry, with different plasma-cell thresholds across studies; and (iii) differences in the sampling approach and in microbiological strategies employed (culture versus high-throughput sequencing), which condition the clinical interpretation of findings. Altogether, these factors hamper comparison across series and favor disparate prevalence estimates [1,2,3,4,5].

Active microorganisms have been detected in the healthy endometrium with functional implications during the receptive phase. In this context, microbial dysfunction may alter key metabolic pathways for endometrial receptivity. Interest in the endometrial microbiota has grown with the proposal that a *Lactobacillus*-dominated environment may be associated with improved reproductive outcomes, whereas dysbiotic profiles (enriched in facultative and strict anaerobes such as *Gardnerella* spp.) are linked to subclinical inflammation and impaired receptivity. In particular, a higher relative abundance of *Lactobacillus crispatus* versus non-dominated communities has been associated with increased likelihood of conception/live birth across cohorts, whereas less stable configurations (e.g., *Lactobacillus iners* predominance or mixed anaerobic communities) are related to dysbiosis. These observations reinforce the functional relevance of *Lactobacillus* spp. as a marker of homeostasis and a potential modulator of implantation [1,6,7].

However, the low-biomass nature of the endometrium and frequent negative cultures hinder robust characterization of the microbial ecosystem and putative pathogens. Intensified culturomics—an expanded panel of media, atmospheres (aerobiosis/anaerobiosis/CO_2_), enrichment steps, and MALDI-TOF—broadens the spectrum of recoverable microorganisms by combining multiple media, atmospheres, and incubation times; nevertheless, it may underestimate fastidious taxa or organisms in a viable-but-non-culturable state. In parallel, 16S rDNA gene sequencing (NGS) captures greater diversity and detects hard-to-culture taxa, but it is prone to contamination and often offers limited species-level resolution within certain clades, especially in low-biomass matrices. The “intensified culturomics” strategy is proposed to improve recovery of viable taxa and obtain isolates for susceptibility testing, whereas NGS provides a community-level view and captures difficult-to-culture microorganisms; thus, the two techniques are conceived as complementary. This integrative approach aligns with the literature linking CE to adverse reproductive outcomes and with the need for uniform diagnostic criteria [6,7,8].

Within this framework, our group published a narrative synthesis reviewing links between CE, microbiota, and fertility, emphasizing the need for prospective designs and combined methodologies to interpret method-related discrepancies in clinically meaningful terms [7].

Given the growing interest in the role of the endometrial microbiome in reproductive health, the present study aimed to develop a detection protocol for microorganisms in endometrial tissue by combining intensified culturomics and NGS. To this end, we designed and optimized a workflow for bacterial DNA extraction and library preparation from endometrial tissue. NGS results were compared with those from intensified culture to evaluate between-method concordance, identify discrepancies, and assess complementary utility. We additionally analyzed the clinical relevance of microorganisms identified by both techniques, exploring their potential relationship with the reproductive prognosis of infertile women.

## 2. Materials and Methods

Study design and population

We conducted a cross-sectional, observational study with descriptive results including patients from the Assisted Reproduction Unit of Hospital Universitario Virgen de las Nieves (Spain) undergoing assisted reproductive treatment (ART) for infertility with clinical and/or ultrasound suspicion of CE. Clinical suspicion of CE included signs such as chronic pelvic pain, postcoital bleeding, dysmenorrhea, and/or intermenstrual bleeding. Ultrasound findings suggestive of CE were a dense, hyperechoic endometrium and/or the presence of micropolyps. Patients were excluded if, within the six weeks prior to sampling, they had taken probiotics or antibiotics, were pregnant, or had experienced infectious episodes. Each patient underwent hysteroscopy; a pathology workup including assessment of plasma cells by hematoxylin–eosin staining and by immunohistochemistry with anti-CD138; and a microbiological culture of the endometrial biopsy based on an intensified culturomics approach. In addition, endometrial biopsy samples were analyzed by bacterial 16S rDNA gene sequencing. Clinical variables are summarized in Appendix A, Table A1.

Clinical data and ethics

Clinical data were extracted from VRepro (NaturalSoft, Granada, Spain) after providing each patient with written information and obtaining written informed consent. Data were pseudonymized by assigning a sample code and processed in accordance with applicable data-protection regulations (GDPR and Spanish LOPDGDD 3/2018).

Sampling

An endometrial biopsy was obtained from each patient by targeted biopsy under direct visualization during the late follicular phase of the menstrual cycle (days 7–13). All samples were collected by the same clinician to avoid variability attributable to the sampling technique.

Diagnosis of chronic endometritis

CE diagnosis was based on hysteroscopic, histologic (anatomic pathology), and/or immunohistochemical (CD138) findings. Samples were classified as CE-positive when hysteroscopy showed a thickened, edematous, and hyperemic endometrial mucosa covered with micropolyps smaller than 1 mm, or when pathology reported more than five plasma cells on hematoxylin–eosin staining or more than five CD138-positive cells by anti-CD138 immunostaining. Samples were considered positive if at least one of these criteria was met.

Intensified culturomics and multiplex PCR

Biopsy samples were homogenized prior to plating by vortexing with zirconia microbeads (0.1 mm; Sigma-Aldrich, St. Louis, MO, USA). Disruption was performed on a benchtop vortex (VT-3000; Labotaq, Madrid, Spain) at maximum speed. One drop of supernatant was inoculated onto the following media and atmospheres: blood agar, chocolate agar, and modified Thayer–Martin agar in a CO_2_-enriched atmosphere at 37 °C; MacConkey agar and mannitol agar under aerobic conditions; and blood agar under anaerobic conditions. Three additional drops of supernatant were inoculated into thioglycolate broth (BD, Madrid, Spain). All media were incubated at 37 °C for 5 days, with a preliminary plate reading at 48 h. Organisms grown on routine cultures were identified using MALDI Biotyper (Bruker Daltonics, Billerica, MA, USA) or MicroScan (Beckman Coulter, Barcelona, Spain). The presence of *Lactobacillus* spp. was reported semiquantitatively as scant, moderate, or abundant based on colony growth across streaked quadrants on solid medium: scant = at least 10 colonies in the first quadrant; moderate = at least 10 colonies in each of the first two quadrants; abundant = at least 10 colonies in each of the first three quadrants. Multiplex PCR was performed on biopsy material to detect specific etiologic agents on the BD MAX platform (Becton Dickinson, Sparks, MD, USA), using BD MAX CT/GC for *Chlamydia trachomatis* and *Neisseria gonorrhoeae* and BIO-GX for *Mycoplasma hominis*, *Mycoplasma genitalium*, *Ureaplasma urealyticum*, and *Ureaplasma parvum.*

16S rDNA gene sequencing (V3–V6)

In parallel, on the same bead-processed, vortexed samples, DNA extraction was performed in a laminar-flow cabinet using the QIAamp UCP Pathogen kit (Qiagen, Hilden, Germany) following the manufacturer’s instructions. DNA concentrations were measured with Qubit (Thermo Fisher Scientific, Waltham, MA, USA) using the dsDNA HS (High Sensitivity) kit. The extracted DNA underwent a target PCR of the hypervariable V3–V6 regions of the bacterial 16S rRNA gene. Fragment analysis of the amplicons was performed on the TapeStation 4200 (Agilent, Santa Clara, CA, USA). Subsequently, index PCR and library normalization were carried out. PCR processing included negative controls. Sequencing was performed on the MiSeq platform (Illumina, San Diego, CA, USA) with V2 nano chemistry (2 × 250 cycles), together with a no template control (NTC). The entire workflow from extracted DNA was conducted with the Microbiota Solution B kit (Arrow Diagnostics S.R.L., Genova, Italy) according to the manufacturer’s instructions.

Bioinformatic processing and quality indicators

An automated pipeline was run on raw FASTQ files using microbAT software (v. 1.1.0, SmartSeq S.R.L., Novara, Italy): quality filtering, paired-end merging, chimera removal, and taxonomic assignment against proprietary, vendor-curated 16S rDNA reference database (SmartSeq S.R.L.), generating per-sample reports. Taxa were reported as relative abundance (RA), applying a threshold of RA ≥ 5%. Alpha diversity was described using the Shannon index (H).

Statistical analysis

Analyses were prespecified as descriptive. No severity scoring or functional correlation analyses were defined a priori. Consequently, no inferential modeling of taxon–severity relationships nor prediction of microbial functions was performed. Statistical analysis and figure generation were carried out in R v4.3.2 (R Foundation for Statistical Computing, Vienna, Austria). The Jaccard index was calculated per sample considering isolates recovered in culture, taxa detected by NGS, and those identified by both methods simultaneously. Figures were exported from R and formatted in Microsoft PowerPoint 365 (Redmond, WA, USA).

## 3. Results

In our study, intensified culturomics and bacterial 16S rDNA gene sequencing were performed on 15 endometrial biopsy samples from patients diagnosed with CE. Overall, 13/15 samples (86.7%) showed growth in culture, whereas 100% yielded NGS findings at RA ≥ 5%. NGS provided additional information beyond culture in 93.4% of samples, and conversely, culture detected microorganisms not identified by NGS in 66.7%. Figure 1 displays the species identified across the 15 samples; the *y*-axis shows culture growth, and the *x*-axis depicts NGS detections with their relative abundances.

In the two samples with no growth in culture, NGS identified at least one taxon (RA ≥ 5%). *Lactobacillus* spp. were detected exclusively by NGS in 26.7% of samples, and *Fannyhessea vaginae* showed the highest mean relative abundance by NGS while failing to grow in culture, underscoring methodological complementarity (Figure 2).

Across the 15 samples, the following taxa were recovered exclusively by culture: *Lactobacillus gasseri*, *Lactobacillus jensenii*, *Peptoniphilus* sp., *Bifidobacterium breve*, *Staphylococcus lugdunensis*, and *Streptococcus agalactiae*. Conversely, only by 16S rDNA NGS were *Fannyhessea vaginae*, *Prevotella* sp., *Sneathia* sp., *Brevibacterium* sp., and *Cutibacterium acnes* detected. Finally, *Lactobacillus crispatus*, *Lactobacillus iners*, and *Gardnerella* sp. were identified by both methods (Figure 3).

The median Shannon index was H = 2.67 (IQR = 0.865), consistent with communities not dominated by a single taxon (Appendix A, Table A2). Regarding NGS quality metrics, the median number of high-quality reads was 25,137 (IQR = 15,171) after confirming rarefaction-curve stabilization, and the mean percentage of reads unclassified at the genus level was 35.8%. NGS–culture agreement, measured by the Jaccard similarity index, showed a mean of 0.11.

## 4. Discussion

Microbial patterns: absence of Lactobacillus dominance.

In this cohort of 15 endometrial biopsies with a diagnosis of CE, none of the samples was *Lactobacillus*-dominated (LD). Moreno et al. proposed an operational cutoff of RA > 90% for *Lactobacillus* spp., associating LD profiles with higher implantation/pregnancy/live birth rates. In contrast, non-LD profiles enriched in anaerobes (e.g., *Gardnerella* sp., *Fannyhessea*/*Atopobium* sp., *Prevotella* sp., *Sneathia* sp.) have been linked to CE and adverse reproductive outcomes. Our data are consistent with that literature and with the hypothesis of an infectious/dysbiotic origin of CE. See Appendix A, Table A2 for taxa and annotations regarding CE, reproductive outcomes, or healthy endometrium [9,10].

Functional context of dominant taxa.

From a functional standpoint, LD communities are generally consistent with an acidic, low-inflammatory endometrial milieu and antagonism of opportunistic anaerobes, aligning with the better reproductive outcomes reported across cohorts. *Lactobacillus crispatus* is consistently linked to epithelial protection via abundant lactic acid, pH lowering, hydrogen-peroxide production and bacteriocins, supporting mucosal homeostasis; by contrast, *Lactobacillus iners* is often observed in transitional, less stable communities with a reduced metabolic repertoire and comparatively weaker protective outputs, features that may facilitate dysbiosis under stressors. In contrast, non-LD, anaerobe-enriched configurations are typically associated with low-grade inflammation and impaired receptivity. As 16S rDNA is a DNA-based approach, these functional considerations should be viewed as supportive context rather than proof of viability or mechanism and interpreted alongside culture and clinical findings [1,6,7,9].

Sampling standardization and cycle-phase control.

All biopsies were obtained by the same clinician, which reduces sampling bias inherent to the transcervical approach (potential carryover of microorganisms of vaginal origin). In addition, sampling was standardized to the late follicular phase because this is the time of lowest pregnancy probability in these actively trying patients, minimizing the risk of interrupting an early gestation and reducing confounding from putative phase-related changes in the endometrial microbiota [11].

Complementary detection windows of culture and 16S-NGS.

The two detection methods—intensified culturomics and 16S rDNA sequencing—yielded partial, non-overlapping profiles. Agreement at the genus level, estimated with the Jaccard index (mean = 0.11), indicates limited overlap and method-specific contributions. NGS added findings in the vast majority of samples and, conversely, culture recovered isolates not observed by NGS, supporting a combined-use approach. While culture enables isolation of viable bacteria and antimicrobial susceptibility testing, it may underdetect fastidious or viable-but-non-culturable organisms and offers limited semiquantitative interpretation. Although 16S-NGS captures greater diversity and hard-to-culture taxa, Altmäe et al. emphasize that DNA-based detection does not necessarily equate to viable bacteria, advocating cautious interpretation and rigorous controls. Moreover, 16S often limits phylogenetic assignment to the genus level, and additional studies are needed to resolve bacterial species. The observed discordances between methods are therefore expected given their different detection windows [12,13].

Alpha diversity and exploratory ecological context.

Alpha diversity was described using the Shannon index (H), which integrates richness and evenness and is less sensitive to very low-abundance taxa in low-biomass matrices. The median Shannon index (H = 2.67) suggests non-dominated communities, consistent with dysbiotic profiles. The higher mean relative abundance of *Fannyhessea* (*Atopobium*) *vaginae* by NGS agrees with its reported association with dysbiosis/CE/implantation failure in the literature. Due to the exploratory nature of the study and the limited sample size, beta diversity and clustering analyses were not conducted, as they would lack statistical robustness. Future studies with larger cohorts and inclusion of control groups will enable meaningful comparative ecological analysis of microbial community structure [9,14].

Sequencing depth and quality control.

The median of 24,041 high-quality reads per sample provided adequate sequencing depth, above commonly used thresholds and comparable to prior endometrial studies; moreover, rarefaction-curve saturation and inclusion of negative controls support the robustness of the profiles obtained. As a quality indicator in a low-biomass matrix, we monitored the percentage of reads unclassified at the genus level, given the limited species-level resolution of 16S. In low-biomass sites such as the endometrium, a relatively high proportion of genus-level unclassified reads is common. This reflects weak signal, intrinsic limitations of 16S for fine-scale taxonomy and sequence similarity among close neighbors, and gaps/redundancies in reference databases (not specific to the female reproductive tract) that preclude confident assignments. In addition, stringent decontamination filters are typically applied in low-biomass work, intentionally increasing the unassigned fraction to minimize false positives. Accordingly, a mean of 35.8% is compatible with the nature of the sample and with good methodological practice, and is reported as a quality metric alongside sequencing depth and blank controls [15,16,17,18,19,20,21].

Contamination control and RA thresholding in low-biomass endometrial 16S analyses.

Because the endometrium is a low-biomass matrix, 16S rDNA sequencing is vulnerable to contamination. Therefore, it is essential to include negative controls, pre-specify a relative abundance threshold to limit technical noise and trace contaminants, and corroborate NGS findings against clinical data and culture isolates for appropriate interpretation. There are no universally validated RA thresholds for the endometrial microbiome. In this study, we set RA ≥ 5% to reduce the likelihood of counting spurious low-level signals typical of low biomass—most reported contaminants occur below ~5% RA. This approach aligns with transparency and quality-control guidance for human microbiome studies (STORMS; RIDE) [15,22,23].

Potential contaminants: *Brevibacterium* sp. and *Cutibacterium acnes*.

Contaminant DNA is ubiquitous in reagents and laboratory environments and varies widely in composition between kits and lots; thus, mere detection by 16S is insufficient to infer pathogenicity without additional clinical and microbiological support. *Brevibacterium* sp. lacks consistent evidence as a component of the endometrial microbiota or as an agent of CE/dysbiosis and is frequently reported as a contaminant in low-biomass studies [24]. Detection of *Cutibacterium acnes* by 16S should be interpreted cautiously. Although *C. acnes* is a recognized opportunistic pathogen, in gynecology and deep pelvic sites its clinical relevance should rest on strict criteria: isolation from a sterile site with prolonged anaerobic incubation; correlation with clinical, imaging, or histology findings; and reasonable exclusion of contamination. This underscores the need not to assume pathogenicity from DNA detection alone, to include blank controls, and to report filtering steps applied in 16S workflows [15,24,25,26].

Methodological and matrix-related constraints.

The sample size (n = 15) limits precision and generalizability. Constraints inherent to both 16S and culture remain, in addition to the challenges of a low-biomass matrix. Future studies should therefore include larger cohorts with CE-negative controls to offset these limitations and, using cross-corroboration of clinical data, hysteroscopy, anatomic pathology, culturomics, and 16S rDNA, identify patterns with diagnostic and prognostic value.

The small sample size, the cross-sectional and descriptive design, and the intrinsic constraints of both culture and 16S in a low-biomass matrix limit precision, generalizability, and the ability to draw robust taxon–severity or treatment-response correlations; moreover, 16S rDNA does not support reliable functional inference. A key limitation is the absence of a CE-negative/healthy control group, which precludes attribution of disease-specific dysbiosis beyond descriptive trends. To temper interpretation, we explicitly use the published literature as a reference baseline and refrain from causal claims. Future studies should include larger cohorts with CE-negative controls, prespecified severity and post-treatment endpoints, multimodal cross-corroboration (clinical data, hysteroscopy, anatomic pathology, culturomics, and 16S), and appropriate functional approaches to enable diagnostic and prognostic inference.

Clinical implications.

In a low-biomass setting such as the endometrium, combining intensified culturomics with 16S rDNA sequencing can increase detection when culture is negative, profile *Lactobacillus*-dominant versus anaerobe-enriched communities, and support individualized management when integrated with clinical and histologic findings (e.g., targeted antimicrobial therapy and post-treatment CD138 reassessment). A pragmatic pathway is culture-first, adding 16S in culture-negative or discordant cases, while avoiding causal inference from 16S alone.

## 5. Conclusions

In a low-biomass matrix such as the endometrium, microbiological assessment requires caution and cross-corroboration. Integrating 16S-NGS with intensified culturomics provides a more comprehensive characterization than either technique alone. Our preliminary data suggest that NGS complements diagnosis in low-biomass settings and facilitates stratification by microbial profiles potentially related to reproductive prognosis. Clinically, a combined, stepwise use of intensified culturomics and 16S rDNA sequencing may complement CE work-up—particularly in culture-negative or inconclusive cases. Clinical interpretation should be integrated with hysteroscopy and anatomic pathology. These findings are exploratory and do not establish diagnostic or prognostic utility; such applications will require validation in larger studies including cohorts with CE-negative controls; recruitment for such a control cohort under the same protocol is underway to enable direct case–control comparisons.

## Figures and Tables

**Figure 1 genes-16-01432-f001:**
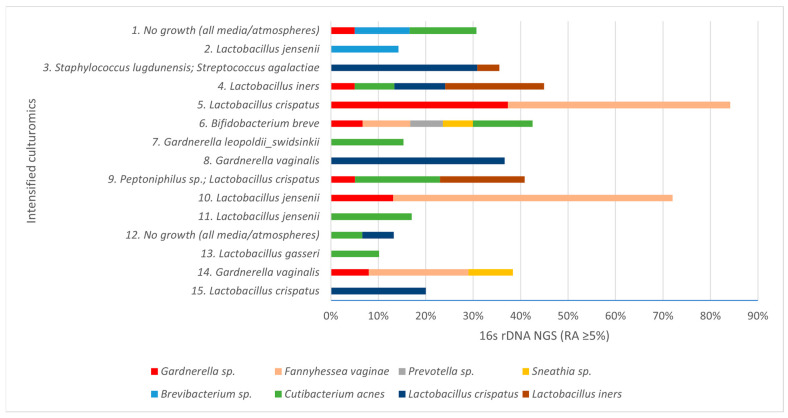
Taxa detected in the samples by culturomics (*y*-axis) and by 16S rRNA gene NGS, with relative abundance (RA, %) on the *x*-axis.

**Figure 2 genes-16-01432-f002:**
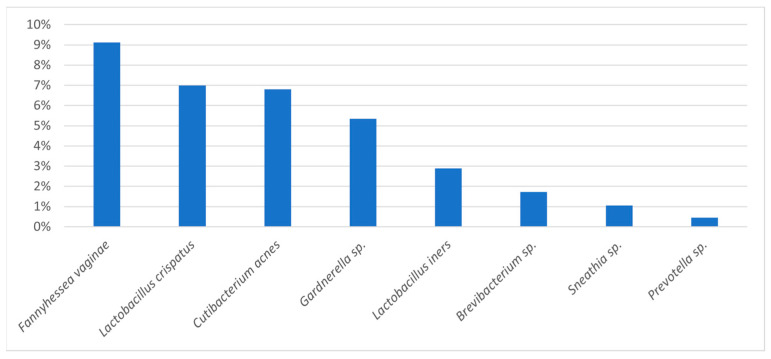
Mean relative abundance of taxa detected by bacterial 16S rDNA gene sequencing.

**Figure 3 genes-16-01432-f003:**
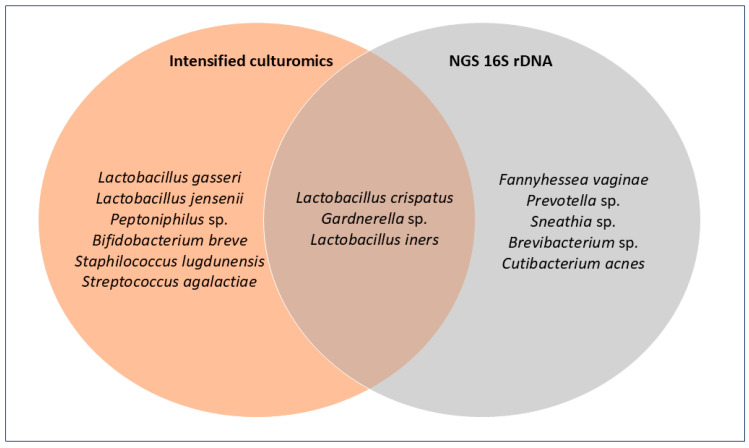
Venn diagram comparing culture and NGS. Each circle contains the species detected by each method; the overlap indicates species identified by both.

## Data Availability

De-identified summary data underlying the figures and tables are provided within the article and Appendix A (Table A1 and Table A2). The raw sequencing reads (FASTQ) and full per-sample reports were not consented for public deposition; therefore, FASTQ files will not be deposited in a public repository. Access to processed data—and, where ethically justified, to raw reads—will be provided in compliance with the General Data Protection Regulation (GDPR) and the Spanish Organic Law on Data Protection and Digital Rights (LOPDGDD) if requested.

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
