# Peer review of "Endometrial Microbial Profile in Infertile Women with Chronic Endometritis: Intensified Culturomics and 16S rDNA Gene Sequencing"

_genes, 2025, doi:10.3390/genes16121432_

Round 1

Reviewer 1 Report

Comments and Suggestions for Authors

This manuscript presents a prospective pilot study comparing intensified culturomics and 16S rRNA gene sequencing for the characterization of endometrial microbiota in infertile women with chronic endometritis (CE). One major comment should be addressed before publication:

The absence of CE-negative or healthy controls significantly weakens the interpretation of microbial differences. Without a reference baseline, it is difficult to distinguish disease-specific dysbiosis from background variability or technical artifacts. 

Minor issue:

  1. The abstract is informative but overly long. It could be condensed to highlight the key findings.
  2. the data availability section should explicitly state whether sequence data (FASTQ files) will be deposited in a public repository (e.g., NCBI SRA)

Author Response

Dear reviewer,

Thank you very much for taking the time to review this manuscript. Please find the detailed responses below and the corresponding revisions/corrections marked in red in the re-submitted files (other colors -blue and green- are used to mark the corrections suggested by other reviewers).

Comments 1: 

This manuscript presents a prospective pilot study comparing intensified culturomics and 16S rRNA gene sequencing for the characterization of endometrial microbiota in infertile women with chronic endometritis (CE). One major comment should be addressed before publication:

The absence of CE-negative or healthy controls significantly weakens the interpretation of microbial differences. Without a reference baseline, it is difficult to distinguish disease-specific dysbiosis from background variability or technical artifacts. 

Response 1: We appreciate this important observation. We agree that the absence of a CE-negative/healthy control group limits disease-specific inference. We have revised the manuscript to make this limitation explicit. Specifically, we added a new paragraph at the end of the Discussion (tenth paragraph, p. 8) clarifying that, in the absence of a CE-negative control group, our findings are interpreted descriptively, contextualized against the published literature as a reference baseline, and that we deliberately focused the present work on the comparative description of two diagnostic microbiological approaches—intensified culturomics and 16S rDNA NGS—highlighting their concordance and complementarity in a low-biomass setting. We also note that a matched CE-negative cohort is currently being recruited under the same standardized workflow. In addition, we added a closing sentence to the Conclusions (p. 9) reiterating this limitation and indicating that the ongoing CE-negative cohort will enable direct case–control comparisons and quantitative inference in future work.

Comments 2: The abstract is informative but overly long. It could be condensed to highlight the key findings.

Response 2: We thank the reviewer for the suggestion to further condense the Abstract. We have shortened it to emphasize study design, key quantitative results, and the main conclusion on method complementarity. We believe this revised Abstract aligns better with the journal’s expectations for brevity and clarity.

Comments 3: the data availability section should explicitly state whether sequence data (FASTQ files) will be deposited in a public repository (e.g., NCBI SRA).

Response 3: Thank you for this clarification request. We have revised the Data Availability Statement to explicitly address sequence deposition. Specifically, we state that FASTQ files will not be deposited in a public repository because patient consent did not cover public release and the PEIBA (Andalusian Biomedical Research Ethics Portal) imposes privacy restrictions. We now detail in the Data Availability Statement section that de-identified summary/processed data are provided within the article and Appendix, and controlled access to processed outputs—and, if scientifically justified, to raw reads—can be arranged upon request and ethics approval, in compliance with GDPR/LOPDGDD.

We sincerely thank the reviewer for the constructive feedback and remain at your disposal for any further clarifications or revisions.

Reviewer 2 Report

Comments and Suggestions for Authors

Dear Authors,

Please find my comments directly on the manuscript.

While the topic is of high interest as infertility rates are increasing, you analysed only CE patients, without a healthy control group. How can anyone attribute your findings to CE and infertility? Maybe all vaginas have these taxa, and you found nothing new and disease associated. A control group is mandatory. Also CE severity could be analyzed in relationship to the taxa abundace and/or presence.

You discuss contamination risk but do not present any analyses of this aspect. Did you have any "blanks"? any analyses to establish a threshold for true positives vs contamination?

Were any beta diversity analyses f clustering analyses conducted?

You cannot talk about the techniques diagnosis capacity since you have only diseased samples.

Statements about adverse outcomes or about diagnostic and prognostic utility are purely speculative as no results exist in the study to back up these statement.

Author Response

Thank you very much for taking the time to review this manuscript. Please find the detailed responses below and the corresponding revisions/corrections marked in blue in the re-submitted files.

Comments 1: Dear Authors,

Please find my comments directly on the manuscript.

Response 1: Thank you very much for reviewing our manuscript and for providing comments directly on the file. We have carefully addressed all your suggestions and implemented the corresponding changes in the manuscript, marking revisions in blue for ease of identification. Please let us know if any further clarifications are needed.

Comments 2: While the topic is of high interest as infertility rates are increasing, you analysed only CE patients, without a healthy control group. How can anyone attribute your findings to CE and infertility? Maybe all vaginas have these taxa, and you found nothing new and disease associated. A control group is mandatory. Also CE severity could be analyzed in relationship to the taxa abundance and/or presence.

Response 2: Thank you for this comment. We agree with yous remarks and, consistent with our replies to the other reviewers, we clarify:

This is a cross-sectional, descriptive study in a low-biomass matrix focused on illustrating complementarity between intensified culturomics and 16S rDNA NGS, not establishing causality. We acknowledge that the absence of a CE-negative/healthy control group and the small sample size (n = 15) limit disease-specific inference. Accordingly, we refrain from causal claims and contextualize our descriptive findings against the published literature on CE and infertility. We did not model associations between taxon abundance and CE severity or treatment response, as such analyses would require CE-negative controls, prespecified endpoints, and appropriate functional approaches. We note that we are currently collecting a CE-negative control cohort under the same standardized workflow to enable direct case–control comparisons in a subsequent study.

Accordingly, we expanded limitations at the end of Discussion (two new paragraphs, p. 8 and 9) and clarified in Methods (Statistical análisis, p. 5) that analyses were prespecified as descriptive, with no inferential taxon–severity/response modeling or functional inference performed. We believe these revisions make the study’s scope and interpretability clearer.

Comments 3: You discuss contamination risk but do not present any analyses of this aspect. Did you have any "blanks"? any analyses to establish a threshold for true positives vs contamination?

Response 3: We agree that contamination control is crucial in low-biomass microbiome studies. As now stated in the Methods (p. 4, lines 172-174), we included negative controls and no template control (NTC) in each sequencing run. No taxa detected in the negative controls appeared above the RA ≥5% reporting threshold in the study samples. We have also clarified in the Discussion (seventh paragraph “Contamination control and RA thresholding in low-biomass endometrial 16S analyses”, p. 8) how this threshold helps minimize noise and avoid false positives in endometrial microbiome analyses.

Comments 4: Were any beta diversity analyses f clustering analyses conducted?

Response 4: Thank you for this suggestion. Given the limited sample size (n = 15) and the descriptive nature of the study, we did not perform beta diversity or clustering analyses, as the statistical power would have been insufficient to support reliable inference. We now explicitly mention this in Discussion (fifth paragraph titled “Alpha diversity and exploratory ecological context”, p. 7) and state that such analyses will be incorporated in the expanded study currently underway.

Comments 5: You cannot talk about the techniques diagnosis capacity since you have only diseased samples.

Statements about adverse outcomes or about diagnostic and prognostic utility are purely speculative as no results exist in the study to back up these statement.

Response 5: We appreciate the reviewer’s point. We would like to clarify that our intention is not to suggest that 16S rDNA-NGS or culturomics provide diagnostic or prognostic value on their own. As stated in the Discussion, this study is descriptive and focuses on the complementarity of both methods in a low-biomass clinical setting, rather than on establishing diagnostic thresholds or clinical prediction. To avoid misinterpretation, we have revised the wording in the Conclusions (p. 9, lines 360-363) to explicitly state that the findings are preliminary and that any potential clinical relevance requires validation in larger, comparative cohorts including CE-negative controls.

Reviewer 3 Report

Comments and Suggestions for Authors

The article is scientifically sound and well-argued. It stands out for integrating culturomics and 16S rDNA sequencing to characterize the endometrial microbiota, offering novelty by revealing complementary yet partially distinct microbial profiles and underscoring their potential clinical relevance in chronic endometritis.

My comments and suggestions.

In the Results section, you could enhance the analysis by including quantitative and functional correlations between specific microbial taxa and the severity of chronic endometritis, as well as the response to treatment.

In the Discussion section, a more detailed exploration of the functional roles of the dominant species would further strengthen the study.

Also, add the limitations of the study at the end of the manuscript, and clinical application.

Author Response

Thank you very much for taking the time to review this manuscript. Please find the detailed responses below and the corresponding revisions/corrections marked in green in the re-submitted files (other colors -blue and red- are used to mark the corrections suggested by other reviewers).

Comments 1: In the Results section, you could enhance the analysis by including quantitative and functional correlations between specific microbial taxa and the severity of chronic endometritis, as well as the response to treatment.

Response 1: We thank the reviewer for this valuable suggestion. Our study was intentionally designed as a cross-sectional, descriptive comparison of intensified culturomics and 16S rDNA NGS in a low-biomass matrix, without prespecified severity indices or functional inference pipelines; therefore, we avoided inferential modeling of taxon–severity or treatment-response correlations to prevent overinterpretation. We have clarified this in Methods (Statistical analysis section, p. 5) and expanded the Discussion (tenth paragraph, p. 8) to explicitly acknowledge this limitation. We also note in the Discussion that future studies should include CE-negative control cohorts, prespecified severity and post-treatment endpoints, and functional analyses to address these questions more definitively.

Comments 2: In the Discussion section, a more detailed exploration of the functional roles of the dominant species would further strengthen the study.

Response 2: We thank the reviewer for this constructive suggestion. While our 16S rDNA approach does not enable direct functional inference, we have expanded the Discussion (second paragraph, p.7) to summarize plausible functional roles of the dominant taxa based on prior literature. The new paragraph integrates these mechanisms as contextual interpretation of our descriptive findings while explicitly avoiding causal claims. Corresponding references have been added.

Comments 3: Also, add the limitations of the study at the end of the manuscript, and clinical application.

Response 3: We thank the reviewer for this helpful suggestion. We have expanded the limitations at the end of the Discussion and added a brief paragraph outlining plausible clinical applications (last paragraph “Clinical implications”, p. 9). Also, conclusions have been updated with a new closing sentence (p. 9, in green) emphasizing cautious, combined use of both methods. These additions clarify the scope of inference and how a combined microbiological approach may be used in practice.

Thank you for your time and thoughtful evaluation of our manuscript. We sincerely appreciate your constructive comments and suggestions, which have helped us clarify the scope, strengthen the Discussion, and improve the overall quality of the work. We hope the revised version satisfactorily addresses your concerns, and we remain at your disposal for any further clarifications.

Round 2

Reviewer 1 Report

Comments and Suggestions for Authors

I am convinced by the revised manuscript. 

Reviewer 2 Report

Comments and Suggestions for Authors

Thank you for the detailed comments and for the revision of your manuscript. There were no blue markings on the manuscript with the modifications as promissed but I managed to find them. Please be more careful next time though and upload also the file with track changes.

Minor comment: In figure 1 and 2 the text is a bit small so I suggest enlarging the figure a bit. You can make use of the white space that is at the left of the page.

Best,

Reviewer 3 Report

Comments and Suggestions for Authors

Accept in present form.